

# Attentional demands associated with augmented visual feedback during quiet standing

Krzysztof Kręcisz[1] and Michał Kuczyński[2]

[1] Faculty of Physical Education and Physiotherapy, Opole University of Technology, Opole, Poland
[2] Faculty of Physiotherapy, University School of Physical Education in Wroclaw, Wroclaw, Poland

## ABSTRACT

To investigate how additional visual feedback (VFB) affects postural stability we compared 20-sec center-of-pressure (COP) recordings in two conditions: without and with the VFB. Seven healthy adult subjects performed 10 trials lasting 20 seconds in each condition. Simultaneously, during all trials the simple auditory reaction time (RT) was measured. Based on the COP data, the following sway parameters were computed: standard deviation (SD), mean speed (MV), sample entropy (SE), and mean power frequency (MPF). The RT was higher in the VFB condition ($p < 0.001$) indicating that this condition was attention demanding. The VFB resulted in decreased SD and increased SE in both the medial-lateral (ML) and anterior-posterior (AP) planes ($p < .001$). These results account for the efficacy of the VFB in stabilizing posture and in producing more irregular COP signals which may be interpreted as higher automaticity and/or larger level of noise in postural control. The MPF was higher during VFB in both planes as was the MV in the AP plane only ($p < 0.001$). The latter data demonstrate higher activity of postural control system that was caused by the availability of the set-point on the screen and the resulting control error which facilitated and sped up postural control.

Corresponding author
Krzysztof Kręcisz, k.krecisz@po.edu.pl

## INTRODUCTION

The contribution of visual control of balance while standing increases significantly with additional visual feedback (VFB) implemented through the conscious control of the center of pressure displacements under the feet (COP) (*Litvinenkova & Hlavacka, 1973*).

A positive effect of VFB has been shown in many studies (*Litvinenkova & Hlavacka, 1973*; *Takeya, 1976*; *Rougier, 2003*) indicating a reduction of the centre of gravity motions with associated increase in muscular activity. However, there are also the investigations that do not confirm such influence. Results of *Danna-Dos-Santos et al. (2008)* indeed demonstrated that participants were unable to decrease sway amplitude when presented with visual feedback, whereas in *Boudrahem & Rougier (2009)* it has been shown that only 69% of subjects were able to use additional visual feedback to reduce COP displacements. Ambiguous findings are also found in reports of attempts to apply VFB technique for rehabilitation. In some studies, the effects appear quite positive (*Shumway-Cook, Anson*

*& Haller, 1988*; *Sihvonen, Sipilä & Era, 2004*; *Cheng et al., 2004*; *Ledebt et al., 2005*; *Sayenko et al., 2010*), whereas others should be considered as scarce (*Walker, Brouwer & Culham, 2000*; *Geiger et al., 2001*). The sources of these discrepancies may be inherent in the way of presenting feedback e.g., insufficient scale display of COP on the screen (*Vuillerme, Bertrand & Pinsault, 2008*) and/or delay of the signal on the screen (*Rougier, 2004*; *Van den Heuvel et al., 2009*), but also high cognitive demands associated with learning a new task (*Wulf & Shea, 2002*; *Van Vliet & Wulf, 2006*). Also, many studies have provided evidence that there are significant attentional requirements for postural control (*Woollacott & Shumway-Cook, 2002*). Further, the attentional demand associated with postural control can be modified by the sensory context (*Vuillerme, Isableu & Nougier, 2006*).

A fuller understanding of postural control mechanisms through the conscious control of the center of pressure displacements under the feet allows for a more in-depth diagnosis of certain pathological conditions, and can also be important in training the balance of both patients and athletes (*Szczepańska-Gieracha et al., 2016*).

Taken together, there is still no consensus as to how vertical posture is controlled when the participants are presented with visual feedback from the actual position of their COP. The predominant view is that attentional resources must be involved due to the larger complexity of the VFB as compared to quiet stance (*Lakhani & Mansfield, 2015*), however little is known in what way these resources are used and whether their shifts facilitate or interfere with maintaining balance. To better elucidate the underlying mechanisms, it seems crucial to compare the COP measures during VFB and quiet stance with simultaneously recorded reaction time task. While the traditional sway measures account for postural performance, of special interest is sway entropy which quantifies the attentional investment in postural control (*Roerdink, Hlavackova & Vuillerme, 2011*).

Therefore, this study aims to answer the question whether and how the postural task with additional visual feedback requires more attentional demands in young adults. Therefore, we examined reaction times during VFB and while changing the amount and structure of COP time series.

## METHODS

Seven young students participated in the study (mean age (SD): 22.9 (1.1) years; range: 22–25 years, three females, four males). All subjects were healthy and did not undergo any disease that might affect the balance system. They gave their written informed consent to the procedure and were naive as to the purpose of the experiment. The study was approved by the Senate Ethics Committee for Research at the University School of Physical Education in Wroclaw. Written informed consent was obtained from all participants.

Data were collected as previously described in *Simoneau, Bégin & Teasdale (2006)* and *Vuillerme, Isableu & Nougier (2006)*. Specifically, postural stability was assessed on a force plate (AccuSway, AMTI, Watertown, MA, USA) in front of a computer monitor positioned at eye level at a distance of 1 m. Participants were asked to perform two different dual tasks. In the reference condition (REF), they were asked to sway as little as possible fixating a white sign on the screen and simultaneously perform a probe-reaction time (RT) task.

The RT task consisted of responding as rapidly as possible to an unpredictable auditory stimulus by pressing a handheld button. Eight reaction time stimuli were presented within each 20 s trial. During the second condition (VFB), the COP position was displayed as a spot (diameter of about 4 mm) on the monitor and the subjects were instructed to keep the spot inside the circle (diameter of about 5 mm) on the monitor and simultaneously perform the RT task. The ratio between the real displacements of the COP and their display on the 19-inch screen was twofold for both the anterior-posterior (AP) and medial-lateral (ML) planes. The AP and ML displacements of the COP were represented on the screen from top to bottom and from left to right, respectively. Postural tasks were the primary tasks, and the subjects were asked to treat it as a priority. Subjects stood in the position as follows: 17 cm between heel centers, with an angle of 14° between the long axes of the feet (*McIlroy & Maki, 1997*). Ten trials for each condition (lasting 20-second each) were presented in pseudo-random (balanced) order.

Data were recorded at sampling frequency 50 Hz. The instantaneous center of foot pressure was calculated from the recorded ground reaction forces in the medial-lateral and anterior-posterior plane separately. The raw COP data were not digitally filtered. Postural balance was evaluated by following parameters based on the COP time-series: standard deviation (SD), mean speed (MV), mean power frequency (MPF) (*Prieto et al., 1996*; *Duarte & Freitas, 2010*) and sample entropy (SE) (*Richman & Randall Moorman, 2000*). The SD and MV measure postural performance, with lower values of these parameters indicating better performance and MPF reveals postural strategy. SD, MV and MPF were computed using MATLAB codes available at *Duarte & Freitas (2010)*. SD is the dispersion of COP displacement from the mean position during a trial duration, MV was calculated as the total COP displacement divided by trial duration and MPF is the mean spectral power frequency of the signal estimated up to 25 Hz range. The power spectral density of the detrended COP data was estimated by the Welch periodogram method. SE indexes the regularity or predictability of a time-series. Increased values of sample entropy, which indicate larger irregularity of the COP, has been attributed to a reduced amount of attention invested in posture (*Roerdink, Hlavackova & Vuillerme, 2011*). Input parameters for estimating the sample entropy were based on the median value of the relative error (*Lake et al., 2002*) resulting in the selection of pattern length $m = 2$ and error tolerance $r = 0.08$ and $0.05$ as optimal parameters for ML and AP time series (normalized to unit variance) respectively of all subjects and trials. SE was computed using MATLAB script available at http://www.physionet.org. RT (in milliseconds) helped for determining the attentional demand associated with postural control and was defined as the time interval between the presentation of the auditory stimulus and the subjects' pressing the handheld button (*Abernethy, 1988*).

A linear mixed-effects model was used to test the effect of VFB on RT and COP indices. Trial, feedback (no feedback vs. additional visual feedback) and their interaction were subjected as fixed factors. The effect of trial was chosen as fixed factor to account for any potential fatigue and/or learning effects. Participants were included as a random intercept to take dependency (correlation structure) in the data into account (*Kuznetsov et al., 2015*; *Boisgontier et al., 2017*). Due to skewed distributions, we used log10-transformed data.

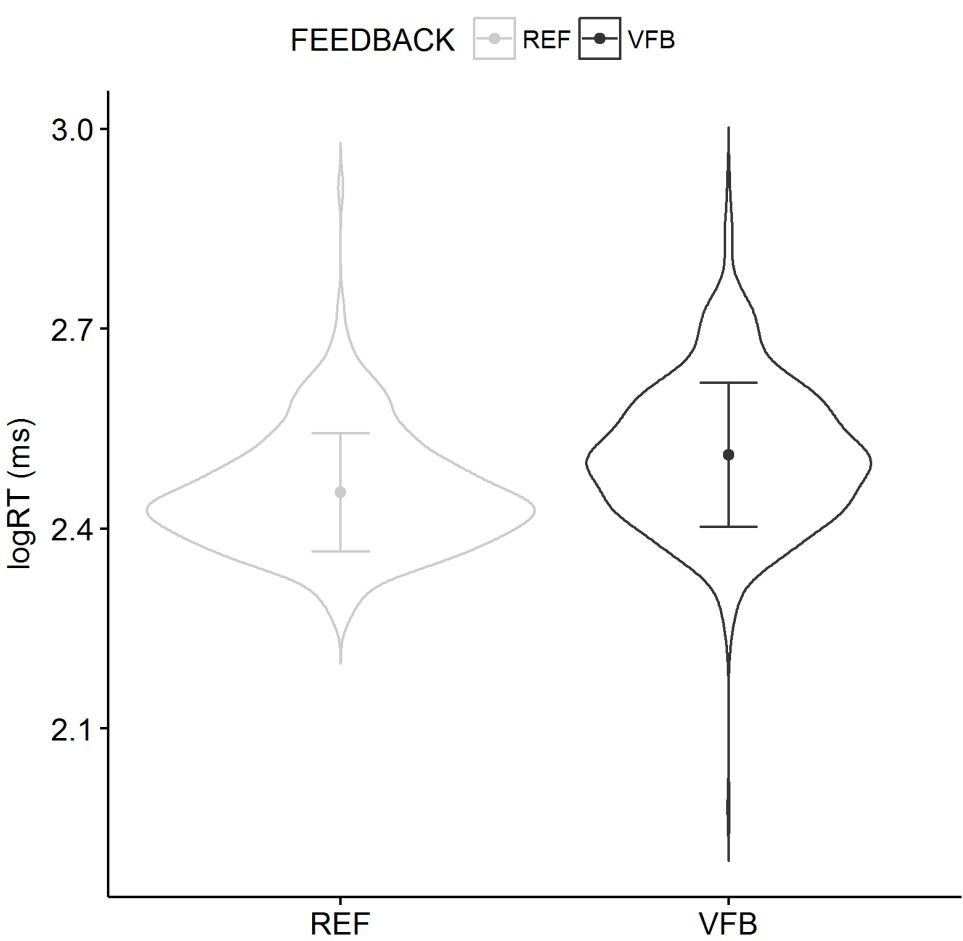

**Figure 1** **Violin plots of the RT for the REF and the VFB conditions collapsed over trials with mean and standard deviations superimposed.** These plots show full distribution of the data obtained by kernel density estimation. The dot symbol denotes mean, whisker denotes standard deviation.

The level of significance was set at $P < 0.05$. Random intercept models take into account the dependence of repeated trials and have substantial advantages over repeated measures ANOVA. All analyses were performed using free and open software JAMOVI 0.8.2.2 with GAMLj module (retrieved from https://www.jamovi.org).

## RESULTS

RT was higher in the VFB condition (fixed effects ANOVA, $F(1, 1094) = 96.89$, $p < 0.001$). No trial main effect ($F(9, 1094) = 1.32$, $p = 0.219$) or feedback × trial interaction ($F(9, 1094) = 1.16$, $p = 0.317$) showed statistical significance. The distributions of the RT are shown in Fig. 1.

For SD there were no interaction feedback × trial effects for both ML ($F(9, 114) = 0.92$, $p = 0.507$) and AP planes ($F(9, 114) = 1.82$, $p = 0.072$). SD was lower in the VFB condition for both ML ($F(1, 114) = 68.24$, $p < 0.001$) and AP planes ($F(1, 114) = 62.47$,

$p < 0.001$); there was no significant main effect of trial (ML: $F(9, 114) = 1.13$, $p = 0.347$; AP: $F(9, 114) = 0.95$, $p = 0.483$).

For MV there were no interaction feedback × trial effects for both ML ($F(9, 114) = 0.96$, $p = 0.479$) and AP planes ($F(9, 114) = 1.40$, $p = 0.198$). MV was higher in the VFB condition for AP plane ($F(1, 114) = 15.08$, $p < 0.001$) and showed no change in the ML plane ($F(1, 114) = 0.71$, $p = 0.402$); there was no significant main effect of trial (ML: $F(9, 114) = 0.85$, $p = 0.570$; AP: $F(9, 114) = 0.74$, $p = 0.675$).

For SE there were no interaction feedback × trial effects for both ML ($F(9, 114) = 0.94$, $p = 0.490$) and AP planes ($F(9, 114) = 1.07$, $p = 0.393$). SE was higher in the VFB condition for both ML ($F(1, 114) = 78.61$, $p < 0.001$) and AP planes ($F(1, 114) = 74.83$, $p < 0.001$); there was no significant main effect of trial (ML: $F(9, 114) = 1.61$, $p = 0.120$; AP: $F(9, 114) = 1.60$, $p = 0.123$).

For MPF there were no interaction feedback × trial effects for both ML ($F(9, 114) = 0.98$, $p = 0.463$) and AP planes ($F(9,114)=0.846$, $p = 0.575$). MPF was higher in the VFB condition for both ML ($F(1, 114) = 68.49$, $p < 0.001$) and AP planes ($F(1, 114) = 99.99$, $p < 0.001$); there was no significant main effect of trial (ML: $F(9, 114) = 0.813$, $p = 0.605$; AP: $F(9, 114) = 0.50$, $p = 0.827$).

The distributions of the COP parameters are shown in Fig. 2.

All analyses are available for download using the Open Science Framework: https://osf.io/mptkr/.

## DISCUSSION

The purpose of this study was to determine whether and how the postural task with additional visual feedback requires additional attentional demands in young adults. The results show that: (1) VFB condition requires additional attentional demands because reaction times were longer, (2) concurrent visual feedback about postural sway shifts focus of attention not directly to postural control because of increase of SE, (3) the implementation of the VFB task has triggered the need for a change in the postural strategy through a reduction in the amount of sway and increase of MV and MPF.

In agreement with our work, the increased COP entropy during visual feedback tasks was reported by *Donker et al. (2008)* and *Lakhani & Mansfield (2015)*. They attributed these results to the effect of using the external reference system which is thought to facilitate more automatic control of posture (*Wulf, McNevin & Shea, 2001*). Similarly, the increased reaction times in our experiments account for shifting the attention of participants to the task of keeping their COP inside the target on the screen. Focusing significant attentional resources on the latter task took from the attention which is normally used to maintain postural control which also implies more automaticity in maintaining balance.

All participants were able to effectively use the VFB in reducing their sway amplitude, yet this activity was accompanied by the increase in sway frequency. Higher frequency of postural sway has been often reported during dual tasks that led to reduced amount of attention which is normally involved in postural control. It is argued that increased sway frequency results from increased joint stiffness (*Vuillerme & Vincent, 2006*; *Bieć et al.,*

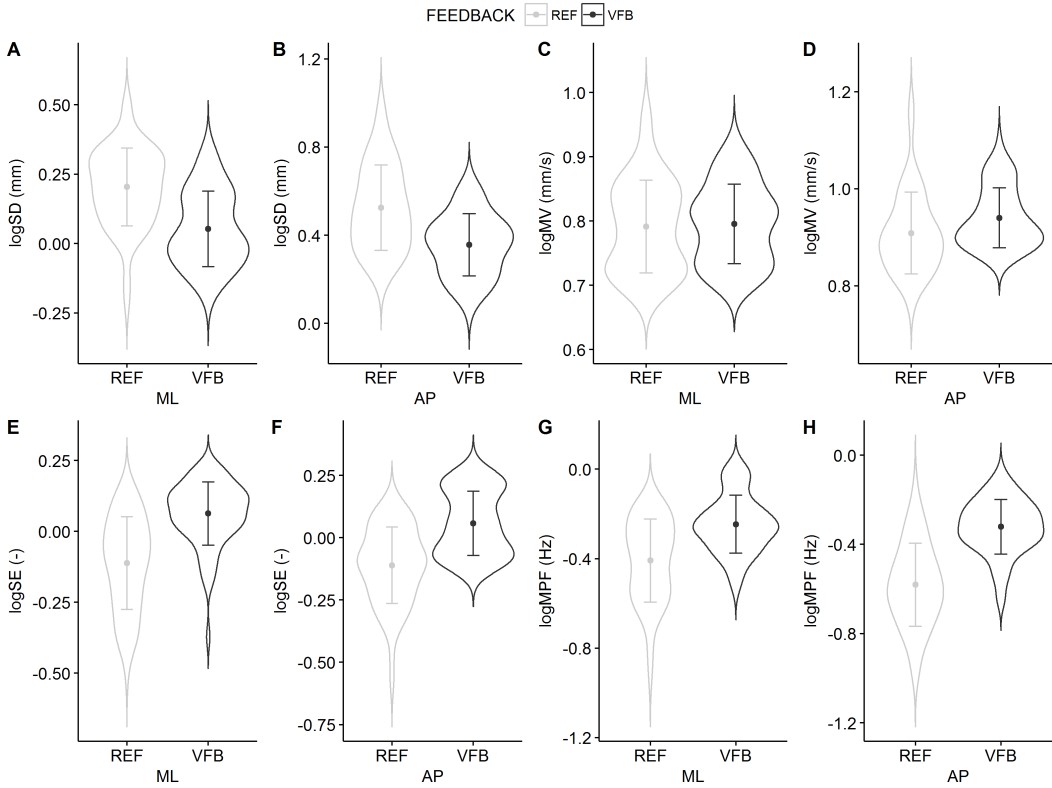

**Figure 2** **Violin plots of the COP parameters for the REF and the VFB conditions collapsed over trials with and mean ± standard deviations superimposed: SD, variability; MV, mean speed; SE, sample entropy; MPF, mean power frequency.** These plots show full distribution of the data obtained by kernel density estimation. The dot symbol denotes mean, whisker denotes standard deviation.

*2014*). Such an interpretation seems justified based on the relationship between the effective postural stiffness and the frequency of the COP signal that was established by *Winter et al. (1998)*. In contrast, *Stins, Roerdink & Beek (2011)* did not find a direct association between postural stiffness and the level of automaticity in controlling posture. However, it is possible that postural tasks with visual feedback have different effects on the distribution of attentional resources than other supplementary cognitive tasks that are apparently not related to posture.

The difference lies in the final application of the attention diverted from posture and invested into the supplementary task. In the latter tasks a necessary portion of attentional resources is being withdrawn from the normal postural control and this loss requires compensation or some other form of reinforcement. According to several authors, this is usually accomplished by promoting the more automatic control process (*Kuczyński, Szymańska & Bieć, 2011*; *Lakhani & Mansfield, 2015*).

However, in the former task, attention transferred to the VFB was indirectly reverted and actually supported the process of postural control. Larger reaction times in the VFB indicate that this task is attention demanding and one can speculate that the diverted attention is necessary for the integration of the ancillary visual input with the remaining

sensory information. Again, elevated sway frequency seems to irrevocably contribute to this purpose. In fact, the stiffening of postural strategy was suggested as the means to perform the postural exploratory and/or monitoring function which significantly increases with the difficulty of postural tasks (*Latash et al., 2003*). This exploratory function of sway is ceaselessly active, even during conscious control of posture, and is thought to have random bearing. A certain level of randomness of spontaneous sway is inherent because of its unconscious origin. However, an additional and quite substantial uncertainty in this signal may develop from the conscious action of participants using the visual error to correct their position inside the target on the screen. Although the purpose of the action is conscious, its timing and magnitude are not, and the two latter factors depend heavily on the sensorimotor abilities and performance of the subjects. In other words, increased sway entropy observed during VFB may not only be the consequence of a more automatic control of posture but also the reflection of increased uncertainty in performance. This would be in agreement with *Morrison, Hong & Newell (2007)* who found that subjects who voluntarily produced random sway motions exhibited higher COP entropy as compared to standing still. In a similar vein *Borg & Laxaback (2010)* postulated that higher entropy may be interpreted as an inability in some circumstances to exert effective attentive control.

## CONCLUSIONS

In conclusion, the VFB is effective in enhancing and improving postural performance. This benefit is associated with increased sensorimotor activity, and its effect on humans, depending on circumstances, may be different. VFB has higher attentional demands as compared to normal stance. VFB increases irregularity and entropy of sway, still presented results seem insufficient to disentangle the role of elevated automaticity or noisiness in these changes.

### Funding
The authors received no funding for this work.

### Competing Interests
The authors declare there are no competing interests.

### Author Contributions
- Krzysztof Kręcisz conceived and designed the experiments, performed the experiments, analyzed the data, contributed reagents/materials/analysis tools, prepared figures and/or tables, authored or reviewed drafts of the paper, approved the final draft.
- Michał Kuczyński conceived and designed the experiments, analyzed the data, contributed reagents/materials/analysis tools, authored or reviewed drafts of the paper, approved the final draft.

## Human Ethics

The following information was supplied relating to ethical approvals (i.e., approving body and any reference numbers):

The Senate Ethics Committee for Research at the University School of Physical Education in Wroclaw granted Ethical approval to carry out the study within its facilities.

## Data Availability

Open Science Framework: https://osf.io/mptkr/.

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
