# Peer review of "Attentional demands associated with augmented visual feedback during quiet standing"

_PeerJ, doi:10.7717/peerj.5101_

## Round 0.1 · original submission · Minor Revisions

As recommended by the reviewers, please re-examine how the raw data is described and presented. Provide more detail on the data capture methods.

Reviewer 1 ·

Basic reporting

Overview
References are appropriate and sufficient background is provided to allow the reader to understand the rationale for the study. The article structure is appropriate. All results relevant to the hypotheses are presented.

Major revisions
1. I believe that the data shared are not sufficiently raw’. The shared data is a database of calculated variables, with one observation per participant per trial. The raw centre of pressure time series files that were used to generate these derived variables should also be shared.

Minor revisions
2. The paper is generally well-written, but there is some unclear language in places; specifically:
a. Page 7, Lines 37-38: unclear what is meant by “the inability of participants to a conscious reduction of body sway”.
b. Page 7, Line 40: I think there is a word missing between ‘Ambiguous’ and ‘are’ – maybe ‘results’/’findings’?
c. Page 7, Line 48: replace ‘evidenced’ with ‘have provided evidence’.
d. Page 11, Line 176: replace ‘significance’ with ‘significantly’.

Experimental design

Overview
The authors compared reaction time and various centre of pressure based measures of balance control between two conditions: no feedback (reference) and with visual feedback of the centre of pressure. The research question is well defined. There is a statement that institutional research ethics approval was received, and that participants provided written informed consent.

Minor revisions
3. Please clarify how many reaction time stimuli were presented within each 20 s trial. Please also comment on the possibility that sway (and consequently COP motion) could have been influenced by the movement required to press the button (independent of any cognitive/attentional influences).
4. Please clarify the size of the items visible on the feedback monitor. On Page 8, Line 75, it seems that the diameter of the circle is 4 pixels. Note that a pixel is not a standard unit of measurement, and the width of a pixel will vary based on the resolution of the screen. Additionally, on most modern monitors, 4 pixels will be very small; is this actually the diameter of the spot rather than the diameter of the circle? A to-scale Figure depicting the visual feedback screen might be helpful.

Validity of the findings

Overview
These findings will be of interest to other researchers in this field.

Minor revisions
5. Data were log transformed prior to analysis. Therefore, I believe the log-transformed data should be presented in the Figures rather than presenting untransformed values. Additionally, since parametric analyses were conducted, please consider presenting means and standard deviations rather than medians and interquartile ranges.
6. In the Results section, please confirm that there were no significant interaction effects (with supporting statistics) prior to presenting main effects. Also, the manner of presentation of the main effects is somewhat confusing and seems to be repetitive. For example, I believe both sentences on Page 9, Lines 119-121 would be clearer written as “SD was lower in the VFB condition for both ML (F(1,114)=68.24, p<0.0001) and AP planes (F(1,114)=62.47, p<0.0001); there was no significant main effect of trial”.
7. Discussion around the sample entropy findings assumes that sample entropy provides information about attentional investment in balance control. While the sample entropy findings are consistent with those of Donker and Lakhani & Mansfield, this finding is somewhat surprising; i.e., one would assume increased attention devoted to balance control when feedback is provided. Perhaps the authors could also speculate on alternative interpretations of sample entropy of COP that would explain previous dual-tasking studies, the feedback studies, and their own findings.
8. Please define MPF in the caption for Figure 2.

Additional comments

No comment.

Reviewer 2 ·

Basic reporting

The article has been written in clear professional English, although there are some minor points that may improve the clarity of your work.
1- At lines 35 and 41, you should explain briefly about some of those positive effects, using the references that you have mentioned.
2- At lines 37 and 38, you have referred to the previous works that have been done by specific authors. You should only put the date in the parenthesis, like so: “Results of Danna-Dos-Santos et al., (2008) indeed demonstrated …”, or “... whereas in Boudrahem & Rougier (2009)…”.
3- At the beginning of line 62, it seems that a verb has been missed.
4- In addition, this will make it clearer if you add SD (standard deviation) in parenthesis after the mean age: “(mean age (SD): 22.9 (1.1) years)”.
5- It seems that some references have been missed, at lines 157, 164, and 169.

Experimental design

1- At lines 36-51 the knowledge gap or shortcoming(s) of previous works have not been clearly pointed out. I suggest that you support the knowledge gap and importance of your work by adding more details about the previous similar studies, this way you will make the importance of your work stand out even more.
2- In the method part of your study, you should add more details about the characteristics of your sample in terms of their sex/gender, height, weight, or their BMI to give your readers a clearer picture of the population of your study.
3- At lines 85 to 102, I suggest that you add a brief description of your signal preparation methods (e.g., the filters that you have used, the method of extracting the frequency content of your COP signals, and the frequency range that you have measured MPF in that range).
4- That would be very helpful, if you could elaborate more on all of the dependent variables, especially about the frequency and entropy measures. For example, you did mention that the MPF reveals postural strategy, but what is exactly the MPF? Or, what is COP-SD?
5- At lines 104 to 110, it is not clearly mentioned that why you have chosen a mixed effect model to analyze your dependent variables. Obviously, visual feedback has two distinct levels (with, and without VFB). In terms of the trial as your other fixed factor, it is not clear that why you have considered it as a categorical variable with different levels. So, I suggest that you elaborate more on that to make it clearer.

Validity of the findings

No comment

Additional comments

The article is an interesting work on the contribution of visual information into the ability of controlling posture, from both linear and non-linear perspectives.
The most important issue is to add more details to the introduction and methods sections of your paper to clarify the gaps, importance of your study, and the methods and measures of your work.

---

## Round 0.2 · accepted · Accept

The reviewers did not indicate any further issues with the manuscript.

#